# Using Background Knowledge from Preceding Studies for Building a Random Forest Prediction Model: A Plasmode Simulation Study

**DOI:** 10.3390/e24060847

**Published:** 2022-06-20

**Authors:** Lorena Hafermann, Nadja Klein, Geraldine Rauch, Michael Kammer, Georg Heinze

**Affiliations:** 1Institute of Biometry and Clinical Epidemiology, Charité–Universitätsmedizin Berlin, Corporate Member of Freie Universität Berlin and Humboldt-Universität zu Berlin, Charitéplatz 1, 10117 Berlin, Germany; lorena.hafermann@charite.de (L.H.); geraldine.rauch@tu-berlin.de (G.R.); 2Chair of Statistics and Data Science, School of Business and Economics, Humboldt-Universität zu Berlin, Unter den Linden 6, 10099 Berlin, Germany; 3Section for Clinical Biometrics, Center for Medical Statistics, Informatics and Intelligent Systems, Medical University of Vienna, Spitalgasse 23, 1090 Vienna, Austria; michael.kammer@meduniwien.ac.at

**Keywords:** calibration, machine learning, sparsity, variable selection

## Abstract

There is an increasing interest in machine learning (ML) algorithms for predicting patient outcomes, as these methods are designed to automatically discover complex data patterns. For example, the random forest (RF) algorithm is designed to identify relevant predictor variables out of a large set of candidates. In addition, researchers may also use external information for variable selection to improve model interpretability and variable selection accuracy, thereby prediction quality. However, it is unclear to which extent, if at all, RF and ML methods may benefit from external information. In this paper, we examine the usefulness of external information from prior variable selection studies that used traditional statistical modeling approaches such as the Lasso, or suboptimal methods such as univariate selection. We conducted a plasmode simulation study based on subsampling a data set from a pharmacoepidemiologic study with nearly 200,000 individuals, two binary outcomes and 1152 candidate predictor (mainly sparse binary) variables. When the scope of candidate predictors was reduced based on external knowledge RF models achieved better calibration, that is, better agreement of predictions and observed outcome rates. However, prediction quality measured by cross-entropy, AUROC or the Brier score did not improve. We recommend appraising the methodological quality of studies that serve as an external information source for future prediction model development.

## 1. Introduction

A central aim for medical researchers is developing fit-for-purpose prediction models of prognosis outcomes [1,2,3]. With recent advances in prediction modeling, researchers now have available a wide range of approaches to choose from that can be grouped into two categories. For example, “data modeling” approaches set up a model structure describing the assumed data generating mechanism, using the observed data to estimate the parameters of that model—most of the traditional statistical modeling approaches belong to this category. In contrast, “algorithmic modeling” does not assume a stochastic data model, and comprises a variety of mostly nonparametric techniques, which use the observed data to find a function of the input to predict the output—many of these methods are referred to as machine learning (ML). The resulting function may be difficult to describe in terms of classical model structures and parameters but is designed to improve prediction accuracy compared to traditional models. This is often achieved using methods such as cross-validation (CV) to balance accuracy of predictions and overfitting [4]. While there are two distinct modeling cultures, there is no sharp distinction between them, and they share the common aim to minimize prediction error (comparing [2] and p. 223, [5]).

These methodology advancements have enabled the estimation of such prediction models in cases where the number of candidate predictor variables exceeds the number of observations. Efficient software is available to perform prediction modeling on large data sets within seconds using model fitting techniques based on data modeling or algorithmic modeling. Models are tailored to allow for direct variable selection (e.g., the Least Absolute Selection and Shrinkage Operator, Lasso, or statistical boosting [6,7]). These approaches select smaller variable subsets, improving the interpretability of results.

The traditional data modeling approach defines a suitable *model structure* that consists of: (1) a set of (candidate) predictor variables, (2) an often linear additive formula that weighs and combines the values of those predictors into a score and (3) an assumed distribution of the outcome given that score. Traditional inference techniques such as p-values or information criteria may then be used to iteratively reduce the set of predictor variables or extend the model with interactions or nonlinear terms [8,9,10]. Naturally, the assumed model structure represents a plausible mechanism that captures the outcome variability as observed in the data. Hence, model development is usually guided by the knowledge of domain experts (i.e., *domain expertise*). Also, the results from previous related studies (i.e., *background knowledge*) guide the selection of candidate predictors and their functional form in the model [11].

Algorithmic modeling approaches such as ML do not require similar assumptions as these approaches are designed to discover patterns from the observed data automatically. As a consequence, it has been argued that ML approaches require more training data to obtain prediction models that are as reliable as traditional modeling approaches [12]. Moreover, even if an ML algorithm is able to identify relevant predictor variables from a larger candidate set automatically, incorporating external information in the selection process may improve interpretability, variable selection accuracy, and consequently, prediction performance. This has been demonstrated using traditional modeling approaches, for instance, Bergerson et al. [13] proposed a weighted Lasso approach combining a penalty with weights based on external information. However, it is unclear to which extent, if at all, nonparametric algorithmic modeling approaches may benefit from using external information derived from prior studies, especially if those studies used traditional statistical modeling approaches, or even applied suboptimal methodology. Therefore, we investigated whether background knowledge generated by traditional modeling approaches, in particular a preselection of the predictor variables performed in previous studies by Lasso regression or by univariate selection, may help inform the design of ML models to obtain more accurate predictions. We focused on methods for binary outcomes in the setting where the number of candidate predictors is large but generally smaller than the number of observations. To investigate the benefit of background information applied with ML methods, we study the widely applied random forest (RF) [4,14].

The remainder of the manuscript is organized as follows. In Section 2, we briefly introduce the prediction methods considered in this paper and outline a typical prognostic research question arising in pharmacoepidemiologic research which motivated our investigation. We describe a large data set connected to the research question which has many common properties of *Big Data*. The data set is characterized by consisting of mainly binary predictor variables with extremely unbalanced distributions and two roughly well-balanced binary outcome variables. This data set will be used as the basis of a complex plasmode simulation, which will be explained subsequently. After reporting the results of our simulation in Section 3, the final Section 4 concludes with a discussion.

## 2. Materials and Methods

### 2.1. Methods to Train Models

#### 2.1.1. Logistic Regression

The logistic regression model predicts the expected value of a binary outcome variable Y∈{0,1} by a linear combination of p predictor variables $X_{1},\ldots ,X_{p}$ as follows: (1)$$\mathrm{Pr}\left(Y=1\right)=expit\left(\beta_{0}+\beta_{1}X_{1}+\ldots +\beta_{p}X_{p}\right)$$ where expit(x)=exp(x)/{1+exp(x)}. The popularity of the logistic regression model may stem from the fact that the regression coefficients $\beta_{1},\ldots ,\beta_{p}$ can be interpreted as the log odds ratios with respect to the occurrence of an event (Y=1) associated with a unit difference in $X_{1},\ldots ,X_{p}$. The parameter $\beta_{0}$ corresponding to a constant $X_{0}=1\;$ serves to calibrate the model such that the sum of model predictions equals the sum of outcome events. The model parameters $\beta_{0},\ldots ,\beta_{p}$ are usually estimated by maximizing the logarithm of the likelihood, that is, the logarithm of the joint probability of the observed outcomes given the predictor variables and the model:$$\ell \left(\beta ;x,\;y\right)=\underset{i}{\sum }y_{i}\log\left({\hat{\pi }}_{i}\right)+\left(1-y_{i}\right)\log\left(1-{\hat{\pi }}_{i}\right),$$
where ${\hat{\pi }}_{i}$ denotes the estimated probability of Y=1 and $y_{i}$ the observed outcome for observation i. Predictions can be obtained by plugging in the estimates of $\beta_{0},\ldots ,\beta_{p}$ and the observed values of the predictors $X_{1},\ldots ,X_{p}$ for an individual into (1). In order to accommodate the nonadditive effects of predictors in the model, one can define further predictor variables as product terms of other predictors. Moreover, the nonlinear effects of continuous predictors can be considered by defining a set of nonlinear transformations of a continuous predictor as further predictor variables. Polynomial transformations, various types of spline bases and fractional polynomials are common choices for this task [11]. To include variable selection into the logistic model, we will contrast the following two approaches.

*Univariate selection:* In real studies, it is not clear up front which variables should be used as predictors. Although not recommended, and despite there being no theoretical justification for it, univariate preselection is still a popular method to reduce the number of variables to include [15,16]. Here, each candidate predictor variable is evaluated in a univariable logistic regression model as the only predictor variable. Those predictors that exhibit a “significant” association with the outcome variable Y are considered for the multivariable model. The significance level α determines how many variables will be included and is often set to 0.05 or 0.20.

*Lasso logistic regression*: In order to avoid overfitting or overparametrization of a model by estimating too many parameters from a data set with a limited number of observations, regularization approaches were developed. One of the most popular approaches is the Lasso which is able to handle regression problems in which the number of predictors approaches or even exceeds the number of observations. Here a penalty term is subtracted from the log-likelihood, equal to a multiple of the sum of absolute regression coefficients, such that the penalized log-likelihood becomes
(2)$$\ell^{*}\left(\beta ;x,\;y\right)=\ell \left(\beta ;x,\;y\right)+\lambda \underset{j}{\sum }\left|\beta_{j}\right|$$

The multiplier λ is a hyperparameter in model fitting which is often optimized by minimizing cross-entropy by CV.

#### 2.1.2. Random Forest

An RF is an ensemble of classification or regression trees in which each tree is grown on a bootstrap resample of the data set [17]. The number of trees in the ensemble is a model parameter. Each tree is constructed recursively: at a given node observations are split into two distinct subsets resulting in two child nodes. This procedure is repeated as long as each node contains a minimum number of observations (in which case it is called a terminal node), constituting a user-chosen parameter of an RF. At each node, the split procedure considers a random subset of the variables in the data set, the size of which is another parameter of the model and is often set to $\sqrt{p}$ (rounded to the next largest integer value), where p is the number of input variables. For each candidate variable, an optimal split point is selected based on a loss function chosen by the user. Among all candidate variables, the one is chosen that optimizes the loss function at its optimal split point. To obtain predictions for an observation the predictions from each tree for that observation are averaged across the ensemble.

While the procedure above is very generic, probability RFs according to Malley et al. [18] were designed to not only provide accurate discrimination but also consistently estimate probabilities of Y=1. In the case of a binary outcome, they use the Gini index measuring node impurity as the splitting criterion, so that the distribution within the child nodes is more homogenous than in the parent node. Furthermore, the minimum node size is set to 10%, and no pruning of the individual trees is performed. A binary prediction in each tree is obtained as a majority vote in the terminal node, and these predictions are then averaged over the whole ensemble to obtain the final probability estimate.

An attractive feature of RFs is the estimation of variable importance using out-of-bag data. Since the variable importance based on the Gini index is known to attribute higher importance to predictors with more splitting points (e.g., continuous variables or categorical variables with more than two categories), alternative approaches, for example, based on permutations of the individual predictors, are preferred [19].

### 2.2. Methods to Evaluate Performance of Model Predictions

In order to evaluate how well a model predicts the outcome, several performance measures are available [20]. Given the model predictions ${\hat{\pi }}_{i}$, their logit transformations ${\hat{\eta }}_{i}=\log\left[{\hat{\pi }}_{i}/\left(1-{\hat{\pi }}_{i}\right)\right]$ and the true outcome values $y_{i}$ in a validation set; i=1,…,N; we considered the following performance measures:Area under the receiver operating characteristic curve (AUROC) or concordance statistic: $E_{i:y_{i}=0,\;j:y_{j}=1}I({\hat{\pi }}_{i}<{\hat{\pi }}_{j})$; ideal value 1Brier score: $E{\left(y_{i}-{\hat{\pi }}_{i}\right)}^{2}$; ideal value 0Calibration slope: slope of a logistic regression of $y_{i}$ on ${\hat{\eta }}_{i}$; ideal value 1Cross-entropy: $-\underset{i}{\sum }[y_{i}\log\left({\hat{\pi }}_{i}\right)+\left(1-y_{i}\right)\log\left(1-{\hat{\pi }}_{i}\right)]$; ideal value 0

### 2.3. Motivating Study

As a motivation for our research, we considered a pharmacoepidemiologic research question in which data on prescriptions of medicines and on hospitalizations were used to prognosticate the adherence to the prescription of a particular blood-pressure-lowering medicine. The research question arose during the conduct of the study of Tian et al. [21] but was not further considered by the authors. In particular, out of a registry of prescriptions and hospitalizations run by the Austrian social insurance institutions, a data set with all new prescriptions of lisinopril, an angiotensin converting enzyme inhibitor, that were filled between 2009 and 2012 in Austria was assembled. Patients were included if they filled a prescription of lisinopril following a wash-out period of at least 180 days. After that index prescription, patients were followed until they filled another prescription of the same substance or until six weeks after the index prescription, whatever happened earlier. Patients in whom the wash-out period of 180 days preceding the index prescription or the outcome assessment window of six weeks following the index prescription were not fully represented in the database were excluded, and also patients who died within 6 weeks from index prescription were excluded. The outcome of interest was treatment discontinuation: patients who did not fill a prescription within six weeks were considered as ”discontinuing treatment” (*Y* = 1), and patients who filled a prescription within 6 weeks as ”continuing” (*Y* = 0). The purpose of the prediction model was to identify patients who are at risk for discontinuation of the lisinopril therapy.

The full data set consisted of 198,895 index prescriptions from 198,895 different patients and 1151 candidate predictors. Demographic descriptors, “recent” prescriptions and hospitalizations (within 14 days before the index prescription) and “previous” prescriptions and hospitalizations (from 180 days to 14 days before index prescription) were available as predictors. Prescriptions were recorded on the basis of the anatomical-therapeutical-chemical classification of level 2 [22] and for hospitalizations, the discharge diagnoses were coded as ICD-10 codes. Moreover, the occurrence of any hospitalization in the ”recent” and ”previous” time windows and the occurrence of any hospitalization longer than 14 days in those windows constituted four further binary variables. A data dictionary for the data set is available in Appendix A. Apart from age in years and year of prescription, all variables were binary and their levels were coded as 0 (absent) and 1 (present). The distribution of the proportions of level 1 for all binary variables is depicted in Figure 1. The majority of binary predictor variables were sparse; the median of their averages was 1/1530, and the first and third quartiles were 1/3371 and 1/112, respectively. The outcome status “discontinuation” occurred with a relative frequency of 0.558.

Fitting an RF with the default settings of the R function ranger::ranger on three-fourths of the data and evaluating predictions in the remaining fourth we obtained an AUROC of 0.646 and a calibration slope of 1.11. Figure 2 and Appendix A show the distribution of Gini variable importance values across the variables.

### 2.4. Setup of the Plasmode Simulation Study

A plasmode simulation study based on the observed properties of the motivating study was performed. The description of the simulation study follows the structured reporting scheme ADEMP as recommended by Morris et al. [23].

#### 2.4.1. Aims

The aim of our plasmode simulation study was to evaluate different strategies to incorporate background knowledge generated from “preceding studies” in a newly developed ML-based prediction model in a “current study” with respect to the performance of the resulting model in a validation set. In particular, we considered several strategies to preselect predictor variables for the prediction model.

#### 2.4.2. Data Generating Mechanisms

Our motivating study was used as the basis of this simulation study. The study data set is considered as the “population” and serves to define training sets for two preceding studies and a current study, and a validation set is used for evaluating the performance of the models developed in the current study. The data set contains the outcome variable treatment discontinuation (*Y*). To also cover situations with a stronger association of the outcome variable with the predictors, we simulated a second outcome variable as follows. First, we divided the complete data set randomly into four approximately equally sized disjoint subsets. Second, we fit an RF using three subsets as the training data set with all available predictor variables. This step was repeated for all four combinations of subsets. Third, we predicted the outcome, that is, we computed the predicted probabilities of discontinuation based on the RF models in the respective remaining fourth of the data set, and transformed the predicted probabilities into log odds. Fourth, we multiplied the log odds with a constant of 3.5 and back-transformed these “reinforced” log odds into “reinforced” probabilities. Fifth, for each observation, we sampled a new binary outcome variable based on Bernoulli distributions with parameters equal to the reinforced probability of that observation. This outcome variable, $Y_{strong}$, was further considered as another target of prediction models. The relative frequency of the status “event” for $Y_{strong}$ was 0.634, and the RF achieved a cross-validated AUROC of 0.801 and a calibration slope of 1.17.

We defined 10 scenarios by varying the sample size (4000, 2000, 1000, 500, 250) and the predictability of the outcome variable (weak or strong). In each scenario, the same sample size was used for the two preceding studies and the current study. The validation set had a fixed size of 10,000 observations and was also redrawn at each iteration of the simulation. Due to the sparsity of most binary predictor variables, they were likely to comprise only a single value in the simulated samples. The likelihood for such degenerate distributions increased with decreasing sample size. Hence the number of useable candidate predictors grew with sample size, which is typical for *Big Data* problems.

We considered 1000 independent replications in each scenario.

#### 2.4.3. Estimands and Other Targets

The estimands in this study were the predictions from the model trained with the data from the “current study” evaluated in the validation set. Moreover, for descriptive purposes, we also recorded the number of predictor variables selected in the preceding studies and the number of non-degenerate predictor variables.

#### 2.4.4. Methods

The first preceding study was analyzed using Lasso logistic regression, selecting the penalty parameter by optimizing the 10-fold cross-validated deviance (the default in cv.glmnet of the R package glmnet [24]). The second preceding study was analyzed by fitting a logistic regression model with all predictors that were significant at a level of 0.05 in univariate logistic regression models.

The current study was analyzed using an RF with five different models (M1–M5) to consider background knowledge, depending on which set of variables was used as input for the RF:All variables (M1, *naïve RF*);Those variables that were selected by the Lasso in the preceding study 1 (M2);Those variables that were selected by the Lasso in preceding study 1 and univariate selection in preceding study 2 (M3);Those variables that were selected by the Lasso in preceding study 1 or univariate selection in preceding study 2 (M4);Those variables that were selected by the “better performing” model. This model was determined by applying the model from preceding study 1 (Lasso) and the model from preceding study 2 (univariate selection) unchanged on the data of the current study and comparing the resulting area under the ROC. The model with the higher AUROC was considered the “better performing” model. (M5)

If the set of input variables contained variables that were degenerate in the current study, then these degenerate variables were not used by the RF. The five RF models were then applied to the validation set to compute the performance measures.

In case Lasso or univariate selection selected no variables, we assumed that background knowledge was not available as if the corresponding preceding studies had never been published. Consequently, we proceeded as follows:If for M2 no variables were selected, we used all candidate predictors instead (no preceding variable selection);If for M3 the intersection was the empty set, we used the result of M4;If for M4 the union of the variable sets was the empty set, we used all predictors;If for M5 one of the models was empty, we used the predictors from the other model as input; if none of the preceding studies selected any predictors we used all predictors in the current study.

#### 2.4.5. Performance Measures

As performance measures, we considered the AUROC, Brier score, calibration slope and cross-entropy introduced in Section 2.2. These measures were computed in each replication and were then summarized by their respective means and standard deviations or, in the case of the calibration slope, with the mean squared values of their logarithms across the replications of each scenario [25]. The latter quantity directly evaluates the deviation of the log calibration slope from its ideal value of zero. For the cross-entropy, we evaluated the maximum contribution in the validation set given by $\underset{i}{\max}\{-[y_{i}\log\left({\hat{\pi }}_{i}\right)+\left(1-y_{i}\right)\log\left(1-{\hat{\pi }}_{i}\right)]\}$.

#### 2.4.6. Pilot Study

The simulation study was first conducted with 100 replications for feasibility, then repeated with 1000 replications.

#### 2.4.7. Software

We used the statistical software R (version 4.0.4) and the following software packages and functions to fit the models:Logistic regression: glm with argument family=binomial(link=logit);Lasso logistic regression: selection of penalty and model fitting with glmnet::cv.glmnet (version 4.1-3, [24]);Random forest: ranger::ranger (version 0.13.1, [14]) with option probability=TRUE to obtain a probability RF.

All remaining parameters were left at their default values.

## 3. Results

### 3.1. Descriptives

#### 3.1.1. Number of Candidate Predictors

Since the candidate predictors had sparse distributions, they were likely to collapse to degenerate distributions in smaller data sets. The mean (minimum, maximum) number of non-degenerate variables for sample sizes 4000, 2000, 1000, 500 and 250 were 966 (918, 1002), 810 (773, 845), 638 (598, 702), 480 (432, 529) and 347 (293, 392), respectively.

#### 3.1.2. Selected Predictors in the Preceding Studies

The number of selected variables by Lasso and by univariate selection depended on the number of candidate predictors but also on sample size. Table 1 provides descriptive statistics.

### 3.2. Comparison of Performance

Under weak predictability, the naive RF, that is, the RF not using any background knowledge (M1), resulted in the best performing models at validation for any sample size (see Table 2, Table 3 and Table 4 for mean cross-entropy, mean AUROC and mean Brier score and Appendix A for the numerical values of mean calibration slope, MSE of log calibration slope, maximum contribution to cross-entropy and standard deviations of all measures).

Under strong predictability, naive RF not using background knowledge was among the optimal methods when evaluating AUROC, Brier score and cross-entropy (Table 2, Table 3 and Table 4). Considering the calibration slope, however, the naive RF produced the worst calibrated models (Figure 3). This was particularly evident when considering the MSE of log calibration slopes, which combines average performance with the variation of performance (Appendix A). For this measure, the RF fueled with predictor sets derived from the union of Lasso and univariate selection performed best at sample size 250. At higher sample sizes, the RF using the intersection of Lasso and univariate selection outperformed all others with one exception at sample size 1000, where method M5, the RF using the preselection based on the “better performing” model (among the Lasso and the univariately selected model trained in the preceding studies) performed equally well.

The non-optimality of the naive RF mainly originated in calibration slopes that were too high, that is, in predictions that were too narrowly scattered around the observed event rate. This behavior attenuated with increasing sample size, but the mean calibration slope was still close to 1.3 at *N* = 4000. With weak predictability, the opposite was observed: at *N* = 250, the mean calibration slope was 0.8, and it reached a value close to 1 at *N* = 4000. In Figure 3, we compared the mean calibration slopes of the naive RF under weak and strong predictability to those of the Lasso models, and to those of the RF using background knowledge in various ways. Unlike the naive RF, mean calibration slopes of the Lasso were fairly constant and independent of sample sizes, yielding values between 1.02 and 1.22. The calibration slopes of the RF using background knowledge moved towards those of the naive RF with increasing sample sizes. This behavior led to an improvement of the calibration slopes at weak predictability but to a paradoxical shift away from 1 with increasing sample sizes at strong predictability. The mean calibration slopes of the RF based on the intersection of selected sets (M3) at weak predictability and smaller sample sizes were impeded by highly influential points, as also suggested by the high standard deviations of that measure (Appendix A).

## 4. Discussion

This plasmode simulation study evaluated if RFs benefit from the preselection of predictors from previous studies that employed a different modeling method. In the simulation, we varied two main drivers of predictive accuracy: the actual predictability of the outcome and the sample size. A special feature of our study was that the number of candidate predictors naturally increased with sample size, as the set of candidate predictors mainly consisted of sparse binary variables. While commonly used measures for predictive accuracy like the AUROC or the Brier score did not improve when restricting the set of candidate predictors, RF models achieved better calibration, that is, better agreement of predictions and observed outcome rates, when the scope of candidate predictors was reduced based on the knowledge created in previous studies. This was observed when the outcome was strongly predictable, but not under weak predictability, where the naïve RF was already the optimal method among the comparators and across all performance measures.

It is remarkable that under strong predictability, the calibration of the naïve RF could be improved by preselection given that the preselection was not based on measures compatible with the nonparametric nature of RFs, but rather based on a linear predictor model fitted with the Lasso, with or without combination with univariate selection. At smaller sample sizes, the RF benefitted from more generously sized candidate sets, while with larger samples, it performed best when based on smaller sets defined as the intersection of Lasso and univariate selection. Interestingly, enriching the candidate set or even blending the Lasso selection with results from univariate selections did not severely worsen the RFs. This is in contrast with results from traditional multivariable modeling, where it is known that because of correlations between predictors, univariate association with the outcome is neither a necessary nor a sufficient condition for a predictor to be important in a multivariable context [15]. When training RFs, for each split in each tree, only a random subset of the predictors is considered and the split is performed in the predictor with the strongest univariate association with the outcome. Hence, by its construction, a RF may benefit from choosing from candidate predictors with strong univariate associations.

Under weak predictability, there was no benefit in restricting the candidate set to predictors selected in preceding studies. Probably, the reason for this was the poor performance and high instability of the models from the preceding studies in these scenarios. At weak predictability, a model can hardly safely distinguish between real and irrelevant predictors as the true predictor-outcome associations are heavily overlaid with noise. Hence, when using predictors that were included in models derived in previous studies, one should also pay attention to the predictive performance of those models. Poorly performing models are probably not so relevant to consider.

The AUROC has several shortcomings because of its nonparametric construction and hence should not be used exclusively to evaluate the performance of prediction models. A high AUROC is achieved by a prediction model as soon as the predictions are correctly ordered, but for that ordering the absolute predicted probabilities are irrelevant. Hence, even a prediction model that provides predictions that are uniformly too high or too low, or are too far off or too close to the marginal outcome rate, may yield a high AUROC. Consequently, the AUROC may obscure problems with the calibration-in-the-large as well as calibration problems such as over- or underfit (see Figure 4 for exemplary calibration plots). However, in the application of a model, the actual accuracy of the predicted probability for an individual is important, and not so much whether it is larger or smaller than that of another individual. Also, the Brier score or the cross-entropy cannot reveal local biases of the prediction model as they average or sum up the prediction error (measured by different loss functions) over the observations of a validation set. For example, it is still possible that there are differences in bias and prediction quality between high-risk and low-risk individuals, which are averaged in these scores.

Calibration as a measure of predictive performance has not yet received the attention it deserves in the evaluation of prediction models [26]. In particular, in this study, we focused on the calibration slope rather than on the calibration in the large. The latter measure evaluates if the mean prediction matches the observed frequency of the predicted outcome status, and is of importance particularly if prediction models are transferred to different target populations. This was not the case in our study and miscalibration in this sense was not expected. We concentrated our investigation on the calibration slope instead. Applying an ideally calibrated model for outcome prediction, one can expect that the predicted probability of the outcome of interest for a subject is similar to the actual relative frequency of the outcome status in similar individuals. Adequate calibration is thus related to the local unbiasedness of residuals, which can be easily checked by smoothed scatterplots of residuals against model predictions. Such plots are central diagnostics in traditional regression modeling but seem to have fallen into oblivion in many contemporary validations of prediction models, which are dominated by nonparametric discrimination measures like the AUROC [27,28].

Already, in his seminal paper on RFs [17], Breiman proposed the iterated estimation of a RF in order to reduce the number of candidate predictors and improve the performance of the forest. In particular, he suggested fitting a RF and evaluating the variable importance for each predictor. In a second run, only the most important predictors from the first run are included. However, Breiman suggested iterating the application of a RF within a single data set. In our study, we simulated the preselection of variables in preceding studies, a situation that prognostic modelers are often confronted with in practice if they screen the literature on similar preceding studies. Moreover, we mimicked another typical practical situation that modelers often face: the preselection of variables in preceding studies is often not compatible with the considered modeling algorithm in the current study, and may even lack methodological rigor. While the use of background knowledge for preselecting predictors has been recommended [29], previous studies on the same topic are often a questionable source if conducted with inferior methodological quality or if poorly reported [30,31].

Previous comparative studies that have evaluated performance at different sample sizes, in particular simulation studies, have often focused on sets of predictors that did not differ between sample sizes. Our study mimicked the typical situation when working with real data sets that the larger the sample size, the more potential predictors can be considered because sparse predictors are more likely to have degenerate distributions in smaller samples. As a consequence, there is a natural preselection towards more robust predictors in smaller samples, and hence some quantities such as the Brier score or cross-entropy do not decrease proportionally to sample size as one might expect with a constant predictor set.

We assumed the same sample sizes for the preceding studies and the current study. This limitation kept the numerical results concise, and an extension is probably not necessary. In a real prediction modeling situation, a researcher might not rely on background knowledge from “smaller” studies than the current study. In contrast, if predictor sets have been generated by larger preceding studies, the observed effects on the performance of the RF will almost certainly amplify.

We used observational data as our “population”, and hence we were unable to say which predictors were truly important and which were irrelevant for prediction. We tried to address this limitation by providing variable importance measures for the variables based on the RF fitted on the full data set. Still, since we do not know the data generating mechanism, we are unsure if ideally predictors should be combined additively or by means of more complex models such as those revealed by RFs. Because many predictors were binary and sparse, additivity is perhaps a plausible assumption. Nevertheless, there could be predictors that are optimally combined using logical OR links. In our study, the predictor variables had differing importance values, but one could also consider variations of this data-generating mechanism. First, there could be a group of only a few variables with high importance, while all other variables have low importance. Second, all variables could have very similar importance. In the first scenario, it is expected that preceding studies could also identify these important variables even if they used Lasso or univariate selection. In the main study, this would also be revealed by the RF, even if it was not fueled with the knowledge from the preceding studies. In the second scenario, when all predictors are similarly (un)important, the selection in preceding studies may be quite random and have no relevance for the main study apart from reducing the variable space. However, these conjectures cannot be proved with our study. By creating a second, artificial outcome variable in which the association of the predictors with the outcome was reinforced, we could study how the strength of predictability of the outcome impacts the benefits of using knowledge from preceding studies.

In our simulation, we assumed that preceding and main studies were sampled from the same population. This assumption is not always justified in practice, where the cohorts used in different studies are sampled from different populations, with different distributions and different predictors being important for predictions in those populations. In such cases, results from preceding studies will not be fully transferable to the main study. Similarly, preceding studies’ results will be less informative if based on small samples, as then the distributions of predictors and their associations with the outcome will vary randomly between studies.

In our motivating example, many predictors indicated the prescription of particular substances, and it is plausible that patients with the same etiology may have received different substances depending on the preferences of the prescribers, leading to a negative correlation between the substance indicators. In such situations, the elastic net [32] might be preferred over the Lasso as it does not select only one predictor out of a group of correlated predictors but tends to include several predictors from such a group. This may provide more robustness against model instability. Although we did not include the elastic net as a method performed in preceding studies, our M4 (RF based on the union of Lasso and univariate selection) may be seen as an approximation to it. By ignoring correlations among predictors, univariate selection may select more than a single representative from a group of correlated predictors with similar associations with the outcome. Therefore, its selection path somehow approximates, for example, the elastic net or the group lasso. Under strong predictability, M4 indeed often outperformed M2 (RF based on Lasso selection). Under weak predictability, however, we could not observe any benefits from applying M4, and then we would also not expect any benefits from using the elastic net instead of the Lasso.

Previously, Heinze et al. [10] pointed out the importance of including predictors with proven strong associations with the outcome in regression models irrespective of their observed association with the outcome in the current study. RFs allow the prioritization of predictors in constructing trees, and a useful and practically relevant strategy may be to classify predictors by their previous relevance in prediction models. Predictors that were selected by preceding studies could be prioritized when constructing trees, while predictors that were not previously used in models may be assigned a lower probability to be considered for node splits. For example, in the ranger package the parameters split.select.weights and always.split.variables allow such a distinction between “strong” and “unclear” predictors, by assigning higher weights to predictors assumed to be strong, or requesting to always split these variables. It was out of the scope of this investigation to optimize and derive general recommendations on the use of these parameters to incorporate background knowledge into RF fitting.

## 5. Conclusions

Overall, we recommend the use of background knowledge generated by preceding studies when RFs are considered for developing prediction models. Such knowledge may greatly help to improve the calibration of RF predictions and particularly to avoid underfitting. Even preselection that is not fully compatible with the nonparametric RF construction may be beneficial. However, researchers are advised to critically evaluate the methodology in such preceding studies, to restrict their consideration to studies in which adequate modeling algorithms have been applied, and to focus on studies that could demonstrate a good performance of their proposed model. In particular, in line with Sun, Shook and Kaye [15] and Hafermann et al. [31] we advise against considering only predictors that proved “significant” in univariate selections in a preceding study that could not demonstrate an appropriate predictive performance of the multivariable model.

## Figures and Tables

**Figure 1 entropy-24-00847-f001:**
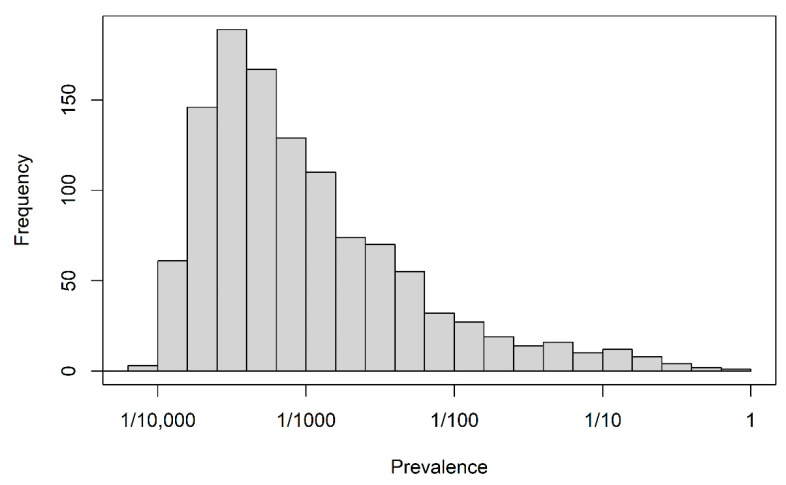
Motivating study: prevalence of 1150 binary predictors.

**Figure 2 entropy-24-00847-f002:**
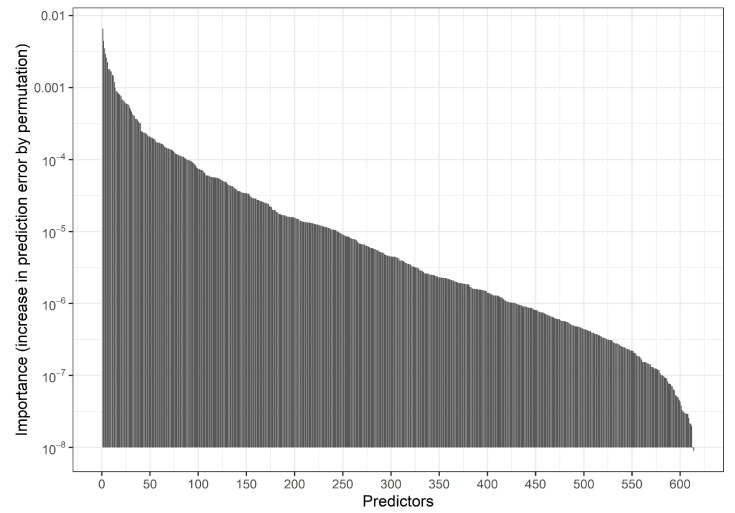
Motivating study: Bar chart indicating the distribution of permutation-based predictor importance values (increase in prediction error by permuting a predictor) in an RF model based on 150,000 individuals. Predictors are ordered by descending importance with more important predictors shown on the left. Permutation predictor importance values were estimated using out-of-bag data in the RF procedure. Here, 537 predictors with negative importance values are not shown. The 20 variables with the highest importance values are provided in Appendix A.

**Figure 3 entropy-24-00847-f003:**
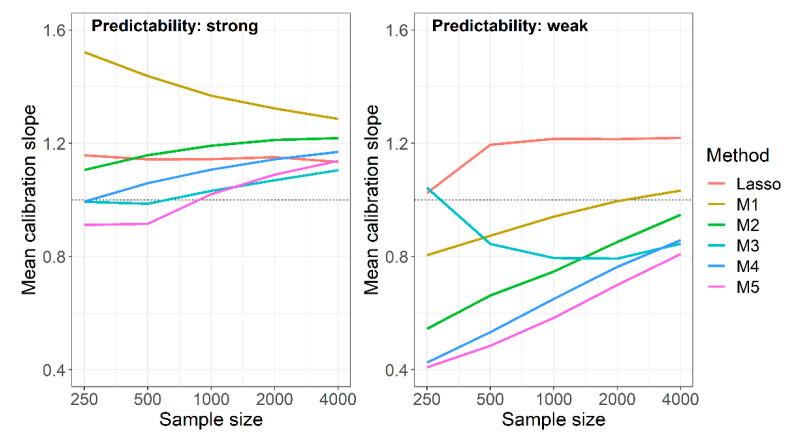
Simulation study, mean calibration slopes achieved by Lasso models and RF models with different preselection of variables in preceding studies for M1: no preselection; M2: preselection based on Lasso; M3: preselection based on intersection of Lasso and univariate selection; M4: preselection based on union of Lasso and univariate selection; M5: preselection based on optimum of the Lasso and univariate model. Results are based on 1000 replications.

**Figure 4 entropy-24-00847-f004:**
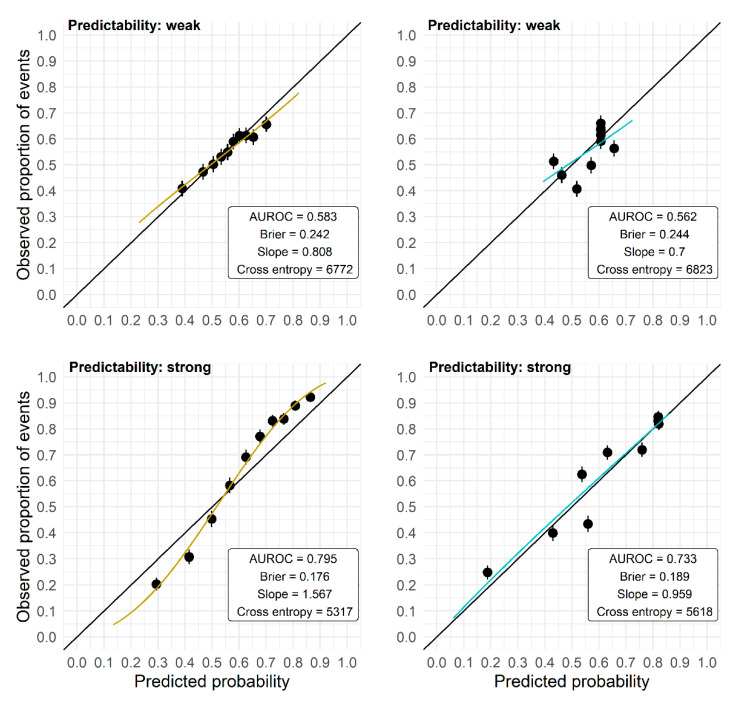
Calibration plots (independent validation) for models developed on two data sets of size *N* = 250 randomly picked from the plasmode simulation. Observed outcome rates in 10 groups defined by the deciles of the predicted probabilities are plotted against the mean predicted probabilities in these groups, and the regression line from a logistic model with the log odds of the predicted probabilities as the only predictor is overlaid. Upper panels: data set with weak predictability, lower panels: data set with strong predictability. M1 (**left**): RF with no preselection; M3 (**right**): RF with preselection based on the intersection of Lasso and univariate selection in preceding studies.

**Table 1 entropy-24-00847-t001:** Simulation study: number of selected variables (mean [range]) in preceding studies. N, sample size. Results are based on 100 replications.

Predictability	N	Lasso	Univariate Selection (α=0.05)	Union of Lasso and Univariate Selection	Intersection of Lasso and Univariate Selection
**Strong**	4000	117 [58, 196]	210 [175, 228]	277 [220, 346]	50 [33, 69]
	2000	103 [47, 178]	111 [105, 118]	179 [129, 246]	35 [22, 51]
	1000	75 [29, 139]	55 [51, 60]	109 [67, 167]	22 [15, 31]
	500	56 [11, 112]	27 [25, 30]	71 [32, 130]	12 [8, 17]
	250	33 [9, 89]	14 [11, 15]	41 [17, 95]	6 [1, 11]
**Weak**	4000	68 [22, 152]	84 [60, 102]	130 [81, 212]	22 [13, 35]
	2000	55 [11, 151]	52 [35, 76]	93 [56, 174]	15 [7, 25]
	1000	37 [0, 131]	30 [17, 45]	60 [29, 144]	8 [0, 18]
	500	21 [0, 88]	17 [4, 29]	34 [4, 96]	3 [0, 10]
	250	14 [0, 76]	8 [0, 15]	20 [2, 81]	1 [0, 7]

**Table 2 entropy-24-00847-t002:** Simulation study: mean cross-entropy achieved by Lasso models and RF models with different preselection of variables in preceding studies for M1: no preselection; M2: preselection based on Lasso; M3: preselection based on intersection of Lasso and univariate selection; M4: preselection based on union of Lasso and univariate selection; M5: preselection based on optimum of the Lasso and univariate model. Results are based on 1000 replications. Bold numbers indicate optimal model in a scenario.

Predictability	Sample Size	Lasso	M1	M2	M3	M4	M5
**Strong**	4000	5081	5050	**4962**	4990	5024	5045
	2000	5152	5116	**5034**	5073	5090	5122
	1000	5253	5195	**5126**	5185	5178	5236
	500	5396	5296	**5255**	5368	5298	5419
	250	5599	**5428**	5455	5609	5466	5581
**Weak**	4000	6621	**6595**	6606	6633	6625	6644
	2000	6660	**6624**	6662	6688	6677	6707
	1000	6713	**6662**	6733	6751	6749	6796
	500	6777	**6711**	6816	6839	6850	6897
	250	6846	**6767**	6894	6957	6959	6987

**Table 3 entropy-24-00847-t003:** Simulation study: mean AUROC achieved by Lasso models RF models with different preselection of variables in preceding studies for M1: no preselection; M2: preselection based on Lasso; M3: preselection based on intersection of Lasso and univariate selection; M4: preselection based on union of Lasso and univariate selection; M5: preselection based on optimum of the Lasso and univariate model. Results are based on 1000 replications. Bold numbers indicate optimal model in a scenario.

Predictability	Sample Size	Lasso	M1	M2	M3	M4	M5
**Strong**	4000	0.809	0.812	**0.818**	0.813	0.811	0.809
	2000	0.803	0.807	**0.812**	0.805	0.805	0.801
	1000	0.794	0.802	**0.803**	0.794	0.797	0.790
	500	0.780	**0.795**	0.789	0.775	0.785	0.773
	250	0.760	**0.785**	0.767	0.744	0.769	0.756
**Weak**	4000	0.627	**0.630**	0.628	0.623	0.623	0.620
	2000	0.619	**0.624**	0.617	0.611	0.614	0.609
	1000	0.606	**0.615**	0.601	0.593	0.603	0.596
	500	0.587	**0.605**	0.579	0.562	0.586	0.577
	250	0.569	**0.592**	0.562	0.547	0.563	0.556

**Table 4 entropy-24-00847-t004:** Simulation study: mean Brier score achieved by Lasso models and RF models with different preselection of variables in preceding studies for M1: no preselection; M2: preselection based on Lasso; M3: preselection based on intersection of Lasso and univariate selection; M4: preselection based on union of Lasso and univariate selection; M5: preselection based on optimum of the Lasso and univariate model. Results are based on 1000 replications. Bold numbers indicate optimal model in a scenario.

Predictability	Sample Size	Lasso	M1	M2	M3	M4	M5
**Strong**	4000	0.167	0.166	**0.163**	0.164	0.165	0.166
	2000	0.170	0.168	0.169	**0.167**	0.168	0.170
	1000	0.174	**0.170**	0.171	0.171	0.171	0.172
	500	0.179	**0.177**	**0.177**	0.178	0.176	0.179
	250	0.188	**0.180**	0.184	0.188	0.182	0.186
**Weak**	4000	0.235	**0.234**	0.235	0.237	0.235	0.236
	2000	0.237	**0.235**	0.237	0.238	0.238	0.239
	1000	0.239	**0.237**	0.240	0.240	0.241	0.244
	500	0.242	**0.239**	0.245	0.245	0.244	0.247
	250	0.245	**0.241**	0.247	0.250	0.250	0.252

## Data Availability

Restrictions apply to the availability of these data. Data were obtained from the Main Association of the Austrian Social Insurance Institutions and are available from the authors with the permission of the Main Association of the Austrian Social Insurance Institutions.

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
