# Peer review of "Using Background Knowledge from Preceding Studies for Building a Random Forest Prediction Model: A Plasmode Simulation Study"

_entropy, 2022, doi:10.3390/e24060847_

Round 1

Reviewer 1 Report

Dear Authors, please find my comments in the attached PDF file.

Author Response

Please find our detailed point-to-point reply attached.

Reviewer 2 Report

In the paper, LASSO has been used to achieve sparsity. To achieve sparsity, LASSO is more applicable but it will not necessarily give the better parameter estimated in the presence of high collinearity. Moreover, LASSO tends to select only one variable among the highly correlated features. It seems to me that the scaled elastic net would be a better pos to use for sparsity.

Regards,

Hayrettin Okut

Author Response

Please find attached the response

Round 2

Reviewer 1 Report

Dear Authors, please find my comments in the attached PDF file.

Author Response

dear referee, please find a point-to-point reply to your comments attached.

Reviewer 2 Report

The current shape of Manuscrıpt is fine and ready to go. Necessary changes have been done appropriately.

Sinceerly

Author Response

Thank you very much for your positive feedback! We are delighted to see that you are happy with our revision.

Round 3

Reviewer 1 Report

The manuscript has improved satisfactorily. I only have one minor revision: The authors use the term "better fitting model" to describe the model that performed better with respect to the AUROC, but a better fitting model is not necessarily a better performing model (keyword: overfitting). Therefore, "better fitting model" should be replaced by "better performing model".

Author Response

please find our response attached.
